# Plant-made polio type 3 stabilized VLPs—a candidate synthetic polio vaccine

Johanna Marsian[1], Helen Fox[2], Mohammad W. Bahar[3], Abhay Kotecha [3], Elizabeth E. Fry[3], David I. Stuart [3,4], Andrew J. Macadam[2], David J. Rowlands [5] & George P. Lomonossoff[1]

Poliovirus (PV) is the causative agent of poliomyelitis, a crippling human disease known since antiquity. PV occurs in two distinct antigenic forms, D and C, of which only the D form elicits a robust neutralizing response. Developing a synthetically produced stabilized virus-like particle (sVLP)-based vaccine with D antigenicity, without the drawbacks of current vaccines, will be a major step towards the final eradication of poliovirus. Such a sVLP would retain the native antigenic conformation and the repetitive structure of the original virus particle, but lack infectious genomic material. In this study, we report the production of synthetically stabilized PV VLPs in plants. Mice carrying the gene for the human PV receptor are protected from wild-type PV when immunized with the plant-made PV sVLPs. Structural analysis of the stabilized mutant at 3.6 Å resolution by cryo-electron microscopy and single-particle reconstruction reveals a structure almost indistinguishable from wild-type PV3.

[1] John Innes Centre, Norwich Research Park, Norwich NR4 7UH, UK. [2] The National Institute for Biological Standards and Control, Potters Bar EN6 3QG, UK. [3] Division of Structural Biology, University of Oxford, The Henry Wellcome Building for Genomic Medicine, Headington, Oxford OX3 7BN, UK. [4] Diamond Light Source, Harwell Science and Innovation Campus, Didcot OX11 0DE, UK. [5] School of Molecular and Cellular Biology, Faculty of Biological Sciences and Astbury Centre for Structural Molecular Biology, University of Leeds, Leeds LS2 9JT, UK. Correspondence and requests for materials should be addressed to G.P.L. (email: George.Lomonossoff@jic.ac.uk)

Poliovirus (PV), a member of the *Picornaviridae*, is the causative agent of poliomyelitis and destroys motor neurons in the central nervous system causing paralysis or even death[1]. The Global Polio Eradication Initiative by the World Health Organization (WHO) has resulted in 99% fewer cases since 1988 by using two very effective vaccines: the live, attenuated oral polio vaccine (OPV), developed by Sabin, or the formaldehyde-inactivated vaccine (IPV), developed by Salk[2, 3]. OPV has been preferred, being relatively inexpensive, easy to administer and additionally inducing gastrointestinal mucosal immunity which prevents virus transmission. However, the genetic instability of OPV means it can regain neurovirulence by reversion mutations which can cause vaccine-associated paralytic poliomyelitis in the vaccine recipient or their contacts, albeit at very low frequency. These attenuated strains may also undergo genetic recombination during replication with other polio or non-polio human enteroviruses, and evolve into circulating vaccine-derived PVs (VDPVs)[4] that display neurovirulent phenotypes similar to wild-type (wt) PVs. OPV also lacks thermostability and requires maintenance of a cold chain. IPV induces effective humoral immunity which protects against disease in the vaccinee but does not always prevent replication of the wt virus encountered subsequently, thus allowing continued transmission within a population[5]. PV strains can be reintroduced into countries that have previously eliminated wild virus circulation via travelers and VDPVs may continue to evolve from the cryptic replication of attenuated OPV strains. Importantly, production of both vaccines requires propagation of large amounts of infectious PV, increasing the risk of an accidental reintroduction. The WHO is therefore seeking novel polio vaccines and our overall objective is to develop recombinantly expressed, stabilized PV virus-like particles (sVLPs) as a cheap and viable source of vaccine to replace current IPVs. Such virus-free vaccines will become increasingly relevant as wt polio is eradicated.

PV occurs in three serotypes (PV1, PV2, and PV3). Its single-stranded (+) sense RNA genome (~7500 nucleotides) contains a single large open-reading frame encoding a 220 kDa polyprotein which is subsequently processed by a series of proteolytic cleavages to generate the viral proteins[6–8]. One of the processing intermediates, P1, is a ~100 kDa protein which is subsequently processed by the viral proteinase 3CD to produce three capsid proteins, VP0–VP3, with the final cleavage of VP0 to give VP2 and VP4 apparently facilitated by the encapsidation of the RNA genome[9]. Mature PV comprises 60 copies each of VP1–4[10, 11] arranged to form an icosahedral capsid 27–30 nm in diameter[12] with distinct antigenic structure, referred to as D (or N (native)) antigen[13], which can conformationally shift to a non-infectious form, called C (or H (heated)) antigen[14]. The C antigen does not elicit a protective immune response and thus is ineffective as a vaccine[15, 16].

A VLP mimics the morphology of a virus particle (ideally retaining the native antigenic conformation) but lacks genomic material and thus can elicit a strong immune response[17–19] but cannot replicate, mutate or recombine. Genome-free (empty) PV particles, such as recombinantly expressed VLPs, do not undergo the final maturation cleavage of VP0 into VP2 and VP4, which rearranges the proteins on the inside of the virus capsid with a concomitant increase in particle stability[9]. Consequently, empty capsids can be easily converted from the D to the C form during extraction and purification[20]. Therefore, to produce an effective vaccine from empty particles, it will be necessary to stabilize the particles to maintain the desired antigenic phenotype. D form enterovirus particles often harbor a long chain fatty acid or "pocket factor" within the core of VP1[21, 22]. By contrast, in the C form this molecule is expelled and there is evidence that tight binding of these host-derived molecules can stabilize the D form and prevent uncoating[23]. Similar small molecule entities have been shown to be useful in stabilizing vaccine antigen[24] and the pocket factor analog GPP3 has been shown to bind PV[25].

In the last 20 years plants have become serious competitors to bacteria, insect cells, yeast or mammalian cells as production systems for pharmaceutical materials. They are robust, inexpensive to grow, bring a low risk of contamination with endotoxins or mammalian pathogens[26–28], and transient expression can be adjusted rapidly with low manufacturing costs[27]. It has been shown that a wide variety of VLP-based vaccine candidates can be efficiently produced in plants, including Medicago Inc.'s influenza vaccine, currently undergoing clinical trials, bluetongue virus, and papilloma-viruses[29–31].

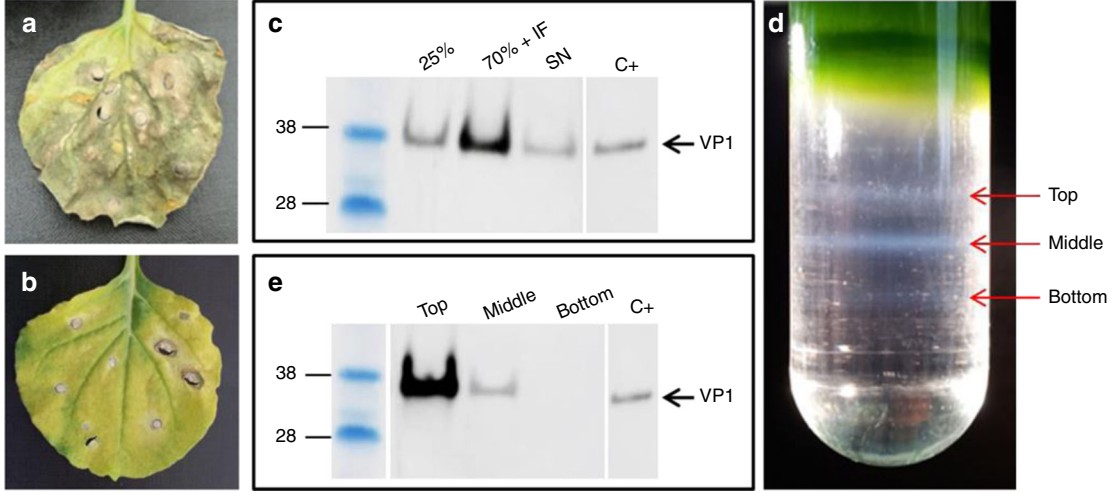

**Fig. 1** Production of PV3 VLPs in plants. **a** *N. benthamiana* leaf 6 days after infiltration expressing PV sVLPs suffering severe necrosis from the high levels of 3CD. **b** Infiltrated *N. benthamiana* leaf 6 days after infiltration expressing PV sVLPs using the downregulated 3CD. **c** Western blot analysis of sucrose cushioned sample showing the strongest signal for VP1 in the 70% sucrose interface (IF). **d** Nycodenz gradient demonstrating the separation of the PV3 SktSC8 VLPs (top band) from contamination. **e** Western blot of Nycodenz gradient confirming that the top *gray band* contains most VP1. The lanes with the pre-stained size markers and the positive control (C+) have been duplicated in **c** and **e** as all the samples were originally analyzed on the same gel (Supplementary Fig. 1)

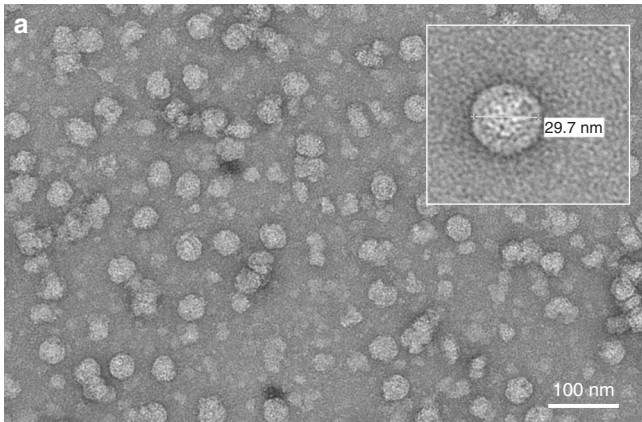

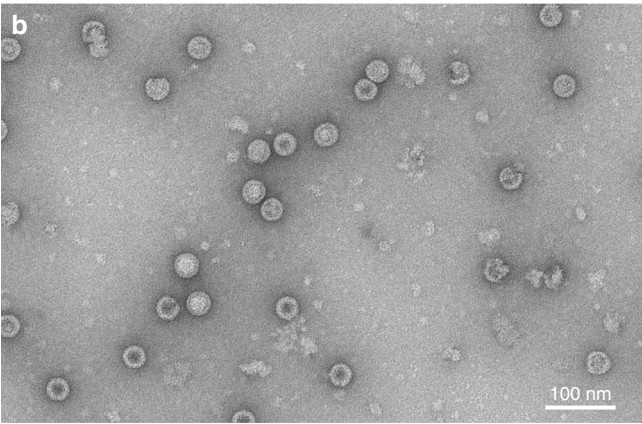

**Fig. 2** Electron microscope analysis of plant-produced PV VLPs.
**a** *N. benthamiana* produced wt PV3 VLPs and **b** PV3 SktSC8 sVLPs
visualized by negative staining and TEM. The inset in **a** represents a higher
magnification image of a particle within the main field

In this paper, we demonstrate expression of PV VLPs of wt
PV3 and the stabilized PV3 mutant SktSC8[32]. We show that this
mutant retains the stability and potency of the current vaccine
IPV and protects mice against challenge with wt PV. Structure
determination of the PV3 SktSC8 sVLPs using cryo-electron
microscopy (cryo-EM) shows that it adopts a native, D, antigenic
conformation akin to wt PVs despite showing little evidence for
bound endogenous pocket factor. These particles do however
bind GPP3 as expected. Furthermore, analysis of the structure
reveals the mechanism of action of the stabilizing mutations.

## Results

**Expression of PV proteins in plants**. To determine if PV VLPs
can be transiently expressed in plants, codon-optimized versions
of the P1 regions of either wt PV3 or the mutant PV3 SktSC8,
identified as having stabilizing mutations within the coat
proteins[32], were inserted into plasmid pEAQ-*HT* to give plasmids
pEAQ-*HT*-PV3 and pEAQ-*HT*-PV3 SktSC8-P1, respectively.
*Agrobacterium tumefaciens* suspensions harboring these
constructs were co-infiltrated with construct pEAQ-*HT*-3CD
to provide proteinase 3CD necessary for processing. While high-
level expression of P1 was tolerated, expression of 3CD caused
severe necrosis in the infiltrated tissue (Fig. 1a). To address this,
3CD expression was downregulated to about 10% of the initial
level by reversing the *HyperTrans* (*HT*)-mutation in the
vector thereby introducing an extra in-frame AUG codon
upstream of the main initiation site[31]. This dramatically reduced
the level of necrosis while still permitting efficient processing of
P1 (Fig. 1b).

**Purification and detection of wt PV3 and PV3 SktSC8 VLPs**.
Leaves were co-infiltrated with pEAQ-*HT*-PV3 and pEAQ-3CD
or with pEAQ-*HT*-PV3 SktSC8-P1 and pEAQ-3CD and har-
vested 6 dpi. The clarified extracts were centrifuged through
sucrose cushions (Fig. 1c) and then through a 20–60% (w/v)
Nycodenz density gradient for further purification. Three gray
bands were visible in the density gradient (Fig. 1d). Each was
retrieved separately and analyzed on a western blot that
revealed that the top band contained the most VP1 (Fig. 1e).
Sodium dodecyl sulfate polyacrylamide gel electrophoresis
(SDS–PAGE) followed by staining with Instant Blue showed
bands consistent with the presence of VP0, VP1, and VP3 in the
preparations, and no evidence for VP2 or VP4.

To confirm particle formation, negatively stained samples
were examined by transmission electron microscopy (TEM).
Figure 2a shows the presence of large amounts of plant-produced
wt PV3 VLPs, although many are irregular in shape or broken.
Regular shaped wt VLPs, as seen in the inset in Fig. 2a, were
rare. By contrast, PV3 SktSC8 proteins assembled into particles of
regular shape and size of 27–30 nm diameter, a morphology
similar to that of native PV (Fig. 2b). The difference in
morphology between the wt and mutant PV3 sVLPs suggests
that the mutations stabilize the particles. The yields in the
purified samples were ~0.06 mg/g fresh weight tissue of PV3
SktSC8 sVLPs and 0.04 mg/g fresh weight of wt PV3 VLPs.

**Structure analysis of PV3 SktSC8 VLPs**. To confirm the
authenticity of the plant-expressed PV VLPs, we have determined
structures of the stabilized plant-expressed PV3 SktSC8 sVLPs by
cryo-EM in the absence and presence of the GPP3 pocket-binding
compound. In total, 4046 particles of PV3 SktSC8 sVLPs and
2060 particles of PV3 SktSC8 sVLPs + GPP3 yielded structures at
3.6 Å and 4.1 Å resolution (Fourier shell correlation (FSC) of
0.143), respectively (Fig. 3, Supplementary Figs 2 and 3). The
models were refined[33] and structure validation was performed by
monitoring real-space correlation and Ramachandran plot
metrics (Supplementary Table 1). The polypeptide main chain
and side chains were well resolved for most of the capsid (Fig. 3d,
e), allowing an atomic model for the majority of the three capsid
proteins (VP0, VP1, and VP3) to be manually built into the cryo-
EM maps. The two particles are almost indistinguishable (rmsd in
Cα atoms 0.72 Å) with both showing D antigen structure—the
final refined atomic models are very similar to the X-ray crystal
structure of the closely related Sabin strain of PV3 (PDB 1PVC),
with only 0.77 Å rmsd in Cα atoms between complete protomers
of the Sabin PV3 (1PVC) and PV3 SktSC8 sVLP. The overall
architecture of the plant-expressed PV3 SktSC8 sVLP preserves
all the structural features characteristic of native conformation
particles including the canyon depression around the
fivefold vertex of the capsid (Fig. 3b). The only significant
changes are on the interior of the particle, where the N-terminal
region of the uncleaved VP0 in the empty sVLPs is less well
ordered than the corresponding regions of VP4 and VP2 in the
mature virus (residues corresponding to VP2 1–11 and VP4 1–25
and 41–69 are not visible in the sVLPs). In addition, the N-
terminus of VP1 from residues 1 to 65 is disordered in the
sVLP structures; such disordering is commonly seen in empty
particles although these are usually in the expanded C antigenic
form. Although the pocket is "open" as expected for a structure
with D antigenicity, very little density is observed for a pocket
factor molecule in the VP1 protein for the PV3 SktSC8 sVLP
structure at 3.6 Å resolution (Fig. 3d), although one or more
pocket factors may be present at very low occupancy. The
structure of the PV3 SktSC8 sVLP mixed with the GPP3
pocket binding molecule determined at 4.1 Å resolution shows

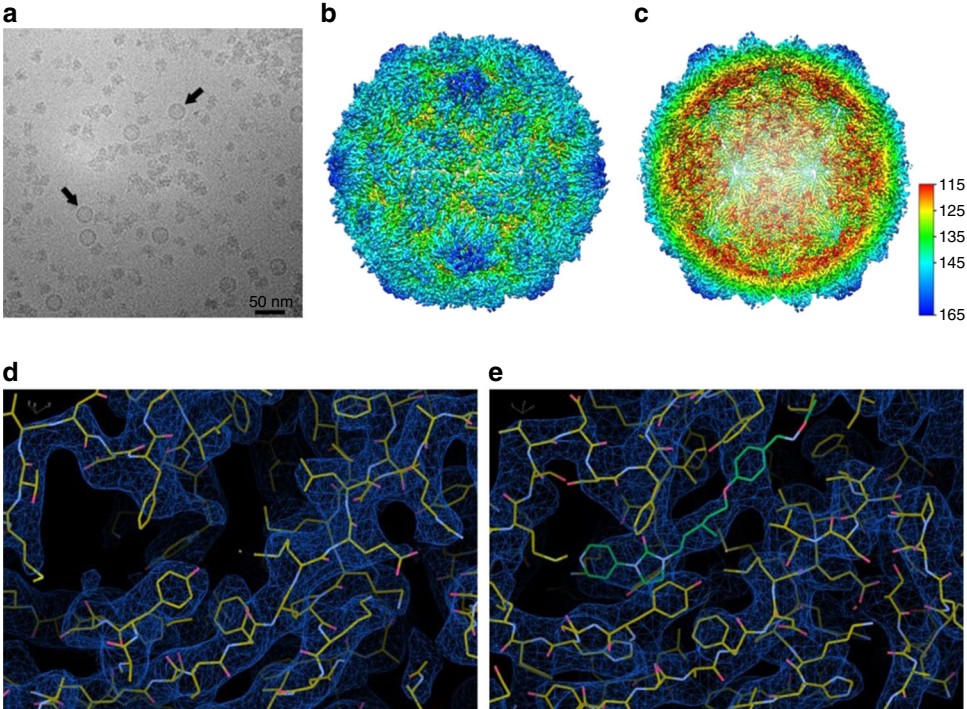

**Fig. 3** Structure determination of PV3 sVLPs. **a** Cryo-EM image of PV3 SktSC8 VLPs in vitreous ice. Examples of some VLPs are highlighted with *black arrows*. Scale bar, 50 nm. **b** 3D reconstruction of PV3 SktSC8 sVLP at 3.6 Å resolution, viewed along the two-fold axis. The surface is colored by radius from the center of the VLP from *blue* through *green* to *red* from the highest to the lowest radius. **c** A central slice through the VLP to show the empty internal surface viewed along the two-fold axis and colored as in **b**. **d** Cryo-EM density map at 3.6 Å resolution of PV3 SktSC8 sVLP looking at the VP1 protein pocket. VP1 is shown as sticks with carbon atoms colored *yellow*, nitrogen atoms in *blue* and oxygen in *red*. **e** Cryo-EM density map at 4.1 Å resolution of the PV3 SktSC8 sVLP with GPP3 pocket factor compound bound to the VP1 protein pocket. VP1 is shown as sticks and colored as in **d**. The GPP3 molecule is colored in *green*. Cryo-EM density maps are contoured at 1.0σ

that although the VP1 pocket is in the same conformation as seen for apo PV3 SktSC8 GPP3 has clearly bound within it (Fig. 3e). The binding mode of GPP3 to VP1 is very similar to structures of pocket factor analogs bound to other D form PV capsids, and to GPP3 bound to enterovirus 71 and Coxsackievirus A16[25, 34], confirming that the plant-expressed PV3 SktSC8 sVLP assembles in an authentic D form competent to bind pocket factor molecules. It is possible that there is a lower concentration of factors suitable to substitute for the usual pocket factor(s) in the cytoplasm of plant cells than mammalian cells, explaining the slightly lower stability observed for plant-produced sVLPs compared to VLPs produced from viral infection of mammalian cells.

Seven out of the eight mutations present in PV3 SktSC8 are visualized in the cryo-EM density maps (the eighth occurring in a region of disordered structure), with clear density consistent with each mutated residue (Fig. 4a). The mutations are distributed evenly across the protomer (Fig. 4b), and are involved in either stabilizing the hydrophobic core of individual protomer subunits (VP1 T105N and VP3 L85F), increasing the hydrophobic packing between protomers within the pentamer of the VLP (VP2 L215M and VP3 H19Y), or, in one case (VP1 F132L), modulating the lining of the VP1 pocket where the pocket factor binds. This pocket factor mutation actually increases the volume of the pocket and there is no evidence that it would stabilize the pocket-factor bound D state; it is therefore possible that it acts instead by destabilizing the C state. For VP2 L18I the change is very small, and it is unclear what effect the change might have, if any, though it is known to have significant biological consequences[21]. VP2 D241E is chemically conserved and it is not clear how the increased length of the side-chain might confer increased

stability. The final change in VP4 occurs in a region disordered in the particle.

**Antigenic analysis and particle stability**. VLPs produced from the wt PV3 construct were entirely of non-immunogenic C antigenic form whereas SktSC8 VLPs, produced in plants or mammalian cells, were predominantly in the native, D antigenic conformation (Fig. 5a). Some C antigen is present in SktSC8 VLP preparations, as it is in all virus preparations and IPV, probably reflecting <100% efficiency in the assembly process. Thus, the stabilizing mutations in PV3 SktSC8 are effective when sVLPs are produced in plants by expression of the capsid precursor and the viral protease as well as in the context of a viral infection of mammalian cells.

Purified wt PV3 (rWT3) VLPs rapidly convert to non-native C antigenic conformation at temperatures above 33 °C[32]. In thermostability assays plant-produced PV3 SktSC8 VLPs (rSktSC8) were much more stable, losing 50% D antigenicity at around 50 °C, similar to type 3 IPV (Fig. 5b). PV3 SktSC8 VLPs (vSktSC8) produced by electroporation of L cells with T7 transcripts were somewhat more stable, possibly reflecting effects of virus gene expression as well as the different cellular environment during particle assembly.

**Immunogenicity studies**. Transgenic mice carrying the human receptor for PV are susceptible to infection and paralysis. Groups of animals were immunized with one or two doses of D antigen corresponding to half a human dose for the sVLPs or IPV. They were then challenged with Saukett PV3. Figure 6 shows the neutralizing antibody titers elicited following

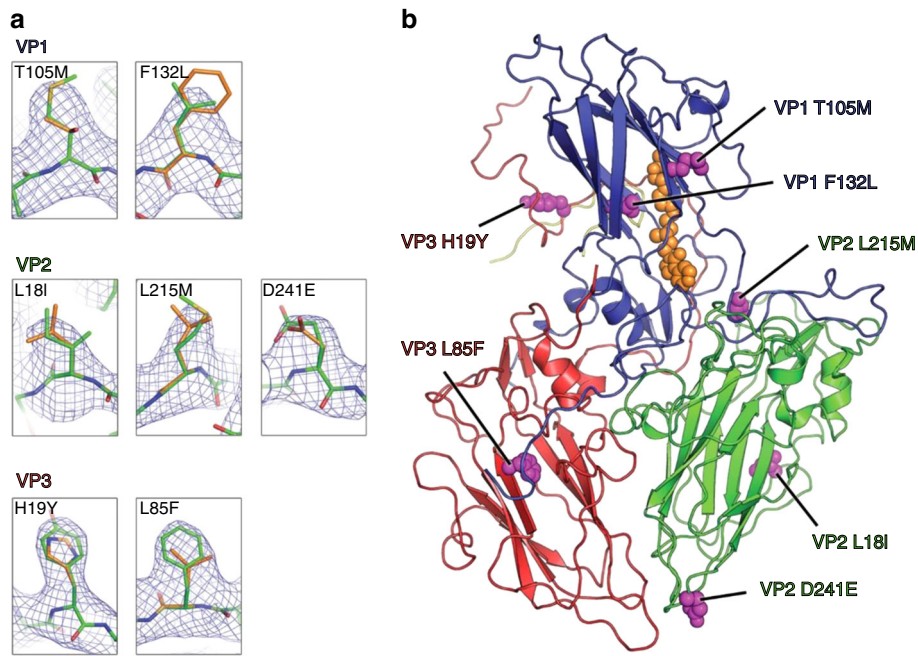

**Fig. 4** Stabilizing mutations in PV3 sVLPs. **a** Mutations in the PV3 SktSC8 sVLP compared to wt sequence. Panels show a snapshot of cryo-EM density for each of the mutations present in the PV3 SktSC8 sVLP. Wt residues are shown as sticks colored *orange*, with mutated residues observed in the 3.6 Å cryo-EM map as *green*. Cryo-EM density maps are contoured at 1.0σ. **b** Cartoon representation of the asymmetric unit for the PV3 SktSC8 sVLP. Individual subunits of the protomer are colored *blue* (VP1), *green* (VP2), *red* (VP3) and *yellow* (VP4). Mutations are shown as spheres and colored *magenta*. The position of the GPP3 molecule observed bound to the VP1 protein pocket of the 4.1 Å structure of PV3 SktSC8 sVLP is shown as *orange spheres*

immunization compared to the pre-challenge levels. The sVLPs from both plants and mammalian cells, given as either one or two doses, induced similar levels of neutralizing antibodies and protection against challenge with live virus as did IPV.

### Discussion
The results of this work show the ability of plants to produce a vaccine candidate against polio. We demonstrate recombinant expression of PV3 sVLPs with a yield of up to 60 mg/kg infiltrated plant tissue. High-level expression of 3CD caused severe damage to the infiltrated leaf tissue, an effect which could be alleviated by reducing expression level. The reduced expression level is still sufficient to permit efficient PV P1 processing resulting in a maximum yield of viral proteins on day 6–7 after infiltration (probed by VP1 expression levels). The processed P1 is able to assemble into VLPs within plants and these can be purified by standard means. When comparing wt PV3 VLPs to PV3 SktSC8 sVLPs by electron microscopy, the structural instability of the wt version becomes apparent; uneven and broken wt particles can be seen, whereas the sVLPs retained typical PV particle morphology. Overall, these findings demonstrate that plants are an effective system for the production of PV VLPs, including stabilized vaccine candidates.

The structure of the PV3 SktSC8 sVLP was solved to 3.6 Å resolution by cryo-EM and showed that the particle is very similar to that of wt PV3 Sabin strain, demonstrating that recombinant plant-expression of PV VLPs can produce authentic particles with the correct antigenic features. The PV3 SktSC8 sVLP structure does not show strong density for a pocket factor in the VP1 pocket. However, it is likely that some VP1 pockets are occupied since the pocket has not collapsed (structure is native antigenicity) and some weak density is observed. It is likely that the VP1 pocket is partly occupied with a heterogeneous mixture of pocket factors, acquired from

the plant cell expression system. Indeed, when the PV3 SktSC8 sVLP was mixed with a well-characterized pocket binding compound (GPP3), a well-occupied complex was seen. The structure of PV3 SktSC8 sVLP + GPP3 showed clear, unambiguous density for the pocket-binding molecule in the VP1 pocket. The structure of PV3 SktSC8 sVLP + GPP3 is unchanged in other aspects compared to PV3 SktSC8 sVLP without GPP3, and both structures had native antigenic conformation. The ability to form a complex with GPP3 with high-occupancy confirms the finding of Rombaut et al.[24] that binding small-molecule pocket-factor mimics provides an additional method for further stabilization of synthetic polio vaccines. This detailed structural analysis provides compelling evidence that authentic, immunologically relevant VLPs can be produced in plants.

The lability of the wt PV3 VLPs was confirmed when analyzing their antigenicity. VLPs produced from the wt PV3 construct were entirely C antigenic whereas PV3 SktSC8 sVLPs were predominantly in the native, immunogenic D antigenic conformation. Immunizing transgenic mice with the plant-expressed PV3 SktSC8 sVLPs induced similar neutralizing antibody responses as IPV and protected animals from challenge with virulent virus, at levels similar to those induced by IPV. These results underline the potential of this plant-produced stabilized mutant as an alternative vaccine against polio.

In conclusion, this study provides proof that the stabilizing mutations in PV3 SktSC8 were effective when the sVLPs were produced in plants by expression of the modified capsid precursor and the viral protease. This mutant is stable in the D form and protects mice when challenged with virulent virus. Cryo-EM analysis revealed that the atomic structure of PV3 SktSC8 sVLP closely resembles the native wt PV. In this study work was focused on PV3; however, stabilized mutants of PV1 and PV2 have also been generated and successfully produced in plants. We report the production of

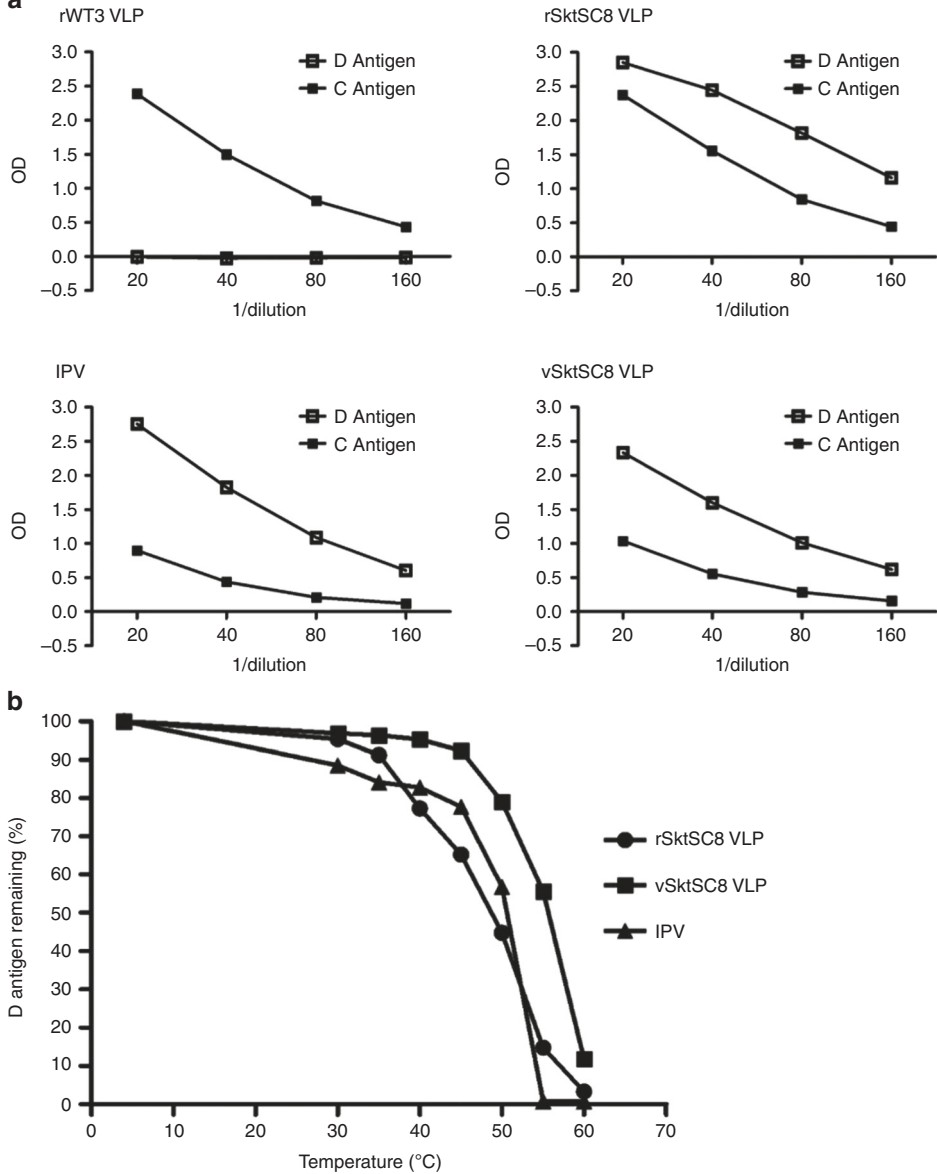

**Fig. 5** Antigenicity of plant-produced PV3 VLPs. **a** Reactivity of VLPs and IPV with MAb 520 (D antigen) and MAb 517 (C antigen) in ELISA. The prefix r denotes recombinantly expressed in plants and v denotes expressed from full-length viral construct following electroporation of L cells. **b** Thermostability of sVLPs. D antigen reactivity of sVLPs and IPV was assessed by ELISA after incubation at different temperatures for 10 min

immunologically effective but non-infectious PV VLPs in plants with the potential to become an alternative vaccine against polio. We also anticipate that the approaches described here could be applied to the production of VLP-based vaccine candidates for other picornaviruses and, indeed, to other families of viruses.

## Methods

**Mutants**. The capsid sequence of the PV3 SktSC8 mutant has been described[32]. Briefly, a full-length PV construct expressing wt 3 Saukett capsid proteins was modified to include eight capsid stabilizing (SC) mutations. Virus preparations were made by high-efficiency electroporation of full-length RNA transcripts into mouse L cells, concentration through 30% (w/v) sucrose cushions and fractionation on sucrose-gradients[32].

**Plasmid constructs**. The PV gene sequences for the 3CD protease, the wt PV3 and the mutant PV3 SktSC8[32] were codon-optimized for *Nicotiana benthamiana* and ordered for synthesis from GeneArt (Life Technologies) with flanking AgeI and XhoI sites. These genes were cloned into separate pEAQ-HT[35] expression vectors producing pEAQ-HT-PV3-P1, pEAQ-HT-PV3 SktSC8-P1, and pEAQ-HT-3CD.

The *HT* mutation of the vector carrying 3CD was subsequently removed to produce the pEAQ-3CD vector. *A. tumefaciens* LBA4404 were transformed with the constructs by electroporation and propagated at 28 °C in Luria-Bertani media containing 50 µg ml$^{-1}$ kanamycin and 50 µg ml$^{-1}$ rifampicin.

**Transient expression**. *A. tumefaciens* containing the respective constructs were grown to stable phase in Luria-Bertani medium supplemented with the appropriate antibiotics. The cultures were then pelleted by centrifugation at 2500 × *g* and re-suspended in MMA buffer (10 mM MES, pH 5.6, 10 mM MgCl$_2$, 100 µM acetosyringone) to an OD$_{600}$ of 0.8 (for pEAQ-*HT*-PV3-SC8Skt-P1) and 0.4 (for pEAQ-3CD). The bacteria were mixed in 1:1 ratio and left at room temperature for 0.5–3 h prior to infiltration. The suspensions were pressure infiltrated into 3-week-old *N. benthamiana* leaves[36]. The infiltrated leaves were harvested 6 days later.

**Extraction and purification**. Infiltrated leaf tissue was weighed and homogenized using a Waring (Torrington, CT) blender with 3× volume of extraction buffer (0.1 M sodium phosphate, pH 7.0) plus added protease inhibitor (Roche, Welwyn Garden City, UK) and then filtered through Miracloth (Calbiochem). The crude extract was centrifuged at 9500 × *g* for 15 min at 4 °C following filtration over

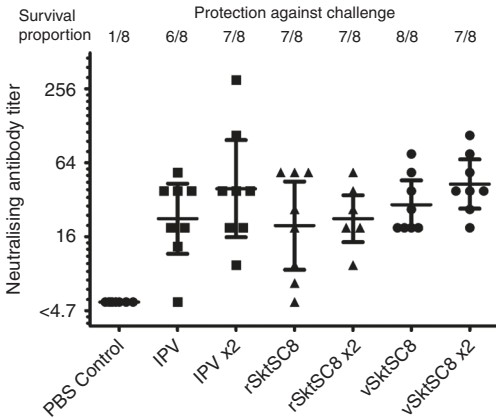

**Fig. 6** Seroconversion and protection against challenge induced by sVLPs. Transgenic mice expressing the human PV receptor were immunized intraperitoneally once or twice (2×) with PBS or 0.5 human dose equivalents of IPV or sVLPs (recombinant—"r" or viral construct—"v" expressed) then challenged intramuscularly with wt virus. Graph shows neutralizing antibody titers against PV3 in blood samples taken the day prior to challenge and survival rates following challenge with wt PV3 Saukett. Bars indicate 95% confidence interval (CI) of the geometric mean titer

a 0.45 μm syringe filter (Sartorius). The clarified extract was then spun through a sucrose cushion (1 ml 70% (w/v) and 5 ml 25% (w/v)) at 167,000 × g for 3 h at 4 °C and the lower fraction retrieved. Sucrose was removed by dialysis using PD10 desalting columns (GE Healthcare) and the volume of the sample reduced using Amicon Centrifugal Filter Units (Millipore). The sample was further purified by centrifugation through a Nycodenz (Axis-Shield) gradient (20–60% (w/v)) in a TH641 ultracentrifuge swing-out rotor (Sorvall) at 247,103 × g for 24 h and 4 °C. VLPs were collected by piercing the side of the tube with a needle and PD10 desalting columns (GE Healthcare) were used to remove the Nycodenz and the samples concentrated using Amicon Centrifugal Filter Units (Millipore).

**SDS-PAGE and western blot analysis**. Protein extracts were analyzed by electrophoresis on 4–12% (w/v) NuPAGE Bis–Tris gels (Life Technologies). Western blot analyses were performed using a monoclonal primary antibody against VP1 protein (Millipore MAB8566) followed by detection with a goat anti-mouse secondary antibody conjugated to horseradish peroxidase, and developed using the chemiluminescent substrate Immobilon Western (Millipore). An uncropped image of the blot is provided in Supplementary Fig. 1.

**Transmission electron microscopy**. VLPs were adsorbed onto plastic and carbon-coated copper grids, washed with several drops of water and then stained with 2% (w/v) uranyl acetate for 15–30 s. Grids were imaged using a FEI Tecnai G2 20 Twin TEM with bottom-mounted digital camera.

**Cryo-EM data collection**. Aliquots of 4 μl of purified PV3 SktSC8 sVLPs were added onto glow-discharged holey carbon copper grids (C-flat, CF-2/1-2 C; Protochips). Grids were blotted for 4 s, in 90% relative humidity, and vitrified in liquid ethane with a plunger device (Vitrobot; FEI). To increase the number of particles in the holes, grids were pre-incubated with 4 μl of sample for 30 s, and unbound sample was removed by blotting with filter paper. For PV3 SktSC8 sVLPs + GPP3, purified PV3 SktSC8 sVLPs were mixed with GPP3 at a ratio of 1 VLP:300 GPP3 molecules and was incubated at 4 °C for 16–18 h overnight. Cryo-EM grids of the PV3 SktSC8 sVLPs + GPP3 complexes were prepared as above.

For PV3 SktSC8 sVLPs cryo-EM data were collected at 300 kV with a Titan Krios microscope (FEI) and a direct electron detector (Falcon II, FEI) at the electron Bio-Imaging Centre (eBIC) at Diamond Light Source. Micrograph images were collected as movies (33 frames, each 0.2 s) and recorded at −2.8 to −1.0 μm underfocus with EPU at a calibrated magnification of ×133,333, thus resulting in a pixel size of 1.05 Å per pixel.

For PV3 SktSC8 sVLPs + GPP3 images were collected at 300 kV using a Tecnai G2 "Polara" microscope (FEI) equipped with an energy filter (GIF Quantum, Gatan) operating in zero-loss mode (0–20 eV energy selecting slit width) and a direct electron detector (K2 Summit, Gatan). Movies (25 frames, each 0.2 s) were collected at a defocus range of −2.8 to −0.8 μm in single-electron counting mode with SerialEM[37] at a calibrated magnification of ×37,037, resulting in a pixel size of 1.35 Å per pixel.

Data collection and refinement statistics are summarized in Supplementary Table 1.

**Image processing and three-dimensional reconstruction**. Similar image processing procedures were employed for the PV3 SktSC8 sVLPs and PV3 SktSC8 sVLPs + GPP3 data sets. Individual frames from each micrograph movie were aligned and averaged using MotionCorr to produce drift-corrected images[38]. VLPs were selected automatically using ETHAN[39] and manually screened using EMAN[40]. Contrast transfer function (CTF) parameters were estimated using CTFFIND3[41] as part of RELION 1.3[42]. Micrographs showing astigmatism or significant drift were discarded and not used for further analysis. All subsequent steps in three-dimensional (3D) reconstruction used RELION 1.3 in accordance with recommended gold-standard refinement procedures[42], and with the application of icosahedral symmetry. For PV3 SktSC8 sVLP, a total of 37,378 particles from 2768 micrographs were subjected to reference-free two-dimensional (2D) class averaging to discard bad particles. The particle population was further improved by 3D classification, using the X-ray structure of PV1 (PDB 1POV) low-pass filtered to 40 Å as an initial model. In total, 4046 particles were selected for 3D auto-refinement. For the PV3 SktSC8 sVLP + GPP3 data set 2060 particles from 1018 micrographs were selected after 2D and 3D classification and 3D auto-refinement used a low-pass (40 Å) filtered model taken from the previously reconstructed PV3 SktSC8 sVLP data set. For both reconstructions the final resolution was assessed using the "gold" standard FSC (FSC = 0.143) criterion[43].

**Atomic model building and refinement**. The X-ray crystal structure of the Sabin PV3 (PDB 1PVC) was manually placed into the cryo-EM density map for the PV3 SktSC8 sVLP data set and rigid-body fitted with UCSF Chimera[44]. The fitting was further improved with real-space refinement using Coot[45]. Manual model building was performed using Coot[45] and in combination with iterative positional and B-factor refinement in real space with Phenix[33]. Only atomic coordinates were refined; the maps were kept constant. Each round of model optimization was guided by cross-correlation between the map and the model. For the PV3 SktSC8 sVLP + GPP3 data set the refined atomic structure of PV3 SktSC8 sVLP was used as a starting model and rigid-body fitted and refined as above. The GPP3 molecule was built into the cryo-EM density within the VP1 pocket using tools within Coot[46]. For both structures, validation was performed with the tools in Coot[45] and the MolProbity[47] functions integrated within Phenix. Refinement statistics are shown in Supplementary Table 1. Molecular graphics were generated using Coot[45], Pymol[48] and UCSF Chimera[49].

**D and C antigen ELISA**. A non-competitive sandwich ELISA assay was used to measure PV D-antigen content[50]. Briefly, two-fold dilutions of antigen were captured with a serotype-specific polyclonal antibody, and then detected using serotype-specific, D antigen (MAb520) or C antigen (MAb517) specific monoclonal antibodies followed by anti-mouse peroxidase conjugate. The D antigen content of each test sample was evaluated against a reference of assigned D antigen content[51] by parallel line analysis (Combistats) in order to derive human-dose equivalents. The human dose for type 3 IPV is 32 D-antigen units.

**Thermostability**. The temperature at which a conformational change from D to C antigenicity occurred was determined by heating at a range of temperatures from 30 to 60 °C followed by D and C antigen ELISA[32]. Samples were diluted in 6-salt PBS to twice the concentration required to obtain an OD of 1.0 in D antigen ELISA, duplicate samples were heated for 10 min at each temperature then diluted 1:1 with 4% (w/v) dried milk in 6-salt PBS and cooled on ice. D and C antigen content was measured by ELISA.

**Immunization challenge**. Transgenic mice expressing the human PV receptor (TgPVR) were immunized and challenged as described previously[32]. TgPVR mice of both sexes (eight per test group) received one or two intraperitoneal injections of PBS (controls) or the equivalent of 0.5 human doses of purified VLPs or the IPV European reference BRP;[51] the second dose, where given, was on day 14. On day 35 blood samples were taken for analysis of neutralizing antibody titers and mice were challenged intramuscularly with the equivalent of 25 times the PD50 of wt PV3 (Saukett) then monitored for any signs of paralysis for 14 days[32].

**Ethical approval**. Animal experiments were performed under licenses granted by the UK Home Office under the Animal (Scientific Procedures) Act 1986 revised 2013 and reviewed by the internal NIBSC Animal Welfare and Ethics Review Board. The TgPVR mouse experiments were performed under Home Office licenses PPL 80/2478 and PPL 80/2050 which were reviewed and approved by the NIBSC Animal Welfare and Ethics Review Board before submission.

**Data availability**. The atomic coordinates of the PV3 SC8 VLP and the PV3 SC8 + GPP3 complex have been submitted to the Protein Data Bank under accession code 5O5B and 5O5P respectively. The cryo-EM density maps of the PV3 SC8 VLP and

the PV3 SC8 + GPP3 complex have been deposited at the Electron Microscopy Data Bank under code EMD-3747 and EMD-3749, respectively. The data that support the findings of this study are available from the corresponding author upon request.

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

## Acknowledgements

At the John Innes Centre (JIC) this work was supported by the UK Biotechnological and Biological Sciences Research Council (BBSRC) Institute Strategic Programme Grant "Understanding and Exploiting Plant and Microbial Secondary Metabolism" (BB/J004596/1) and the John Innes Foundation. The authors thank Jun Dong, Robert Esnouf and Jonathan Diprose for IT support, and the OPIC electron microscopy facility and the staff of the electron Bio-Imaging Centre (eBIC) at Diamond Light Source for microscope provision and technical support. GPP3 was kindly provided by Gerhard Puerstinger, Department of Pharmaceutical Chemistry, University of Innsbruck, Innsbruck, Austria. The work of the Wellcome Trust Centre in Oxford is supported by Wellcome Trust core award 090532/Z/07/Z. The OPIC electron microscopy facility was founded by a Wellcome Trust JIF award (060208/Z/00/Z) and is supported by a Wellcome Trust equipment grant (093305/Z/10/Z). The Wellcome Trust, MRC and BBSRC also support the National EM facility (eBIC), which enabled provision of the K2 detector. M.W.B. and H.F. are supported by a WHO/Gates foundation award (RG.IMCB.I8-TSA-083) and the MRC (grant no. MR/N00065X/1). A.K. supported by Wellcome trust. E.E.F. and D.I.S. are supported by the MRC (grant no. MR/N00065X/1), and D.I.S. is supported as a Jenner Investigator. This work was performed as part of a WHO-funded collaborative effort involving the following Institutions and individuals: University of Leeds: D.J.R., N.J. Stonehouse, C. Nicol, O.O. Adeyemi; University of Oxford: D.I.S., E.E.F., L. de Colibus, M.W.B., C. Porta; University of Reading: I. Jones, M. Uchida, S. Lyons; Pirbright

Laboratory: T. Tuthill, J. Newman; National Institute for Biological Standards and Control: A. Macadam, P. Minor, H. Fox, S. Knowlson; The authors wish to thank Jim Hogle, Ellie Ehrenfeld, and Jeff Almond for support and invaluable scientific input.

## Author contributions
G.P.L., A.J.M., E.E.F., D.J.R., and D.I.S.: Conceived and planned the study. J.M., H.F., M.W.B., and A.K.: Performed experiments and analyzed the data. All authors contributed key ideas, analyzed the data and contributed to writing the paper.

## Additional information

**Competing interests:** G.P.L. declares that he is a named inventor on granted patent WO 29087391 A1, which describes the transient expression system used in this manuscript to express poliovirus VLPs. The remaining authors declare no competing financial interests.

