## [Peer Review File · Nature Communications]

Reviewers' Comments:

Reviewer #1 (Remarks to the Author):

The manuscript "Plant-made polio 3 stabilized VLPs- a candidate synthetic polio vaccine" by Marsian et al. describes the effective assembly of virus-like particles of poliovirus 3 (PV3) with D antigenicity. These plant-expressed VLP were previously stabilized by introducing a few point mutations in the structural capsid proteins crucial for capsid integrity. Analyses by 3D cryo-EM of PV3 VLP at near-atomic resolution are impressive (taking into account the number of particles used for each map) and confirmed 7 of 8 amino acid substitutions (the remaining substitution is located in a disordered region). Finally, immunogenicity studies showed that these PV3 VLP elicit neutralizing anti-D PV3 antibodies. Adapted mice (in which a PV3 receptor is expressed) were immunized, and developed almost full protection to viral challenge. These findings are important for virus-based biotechnology and should be of interest to the structural biology community.

I recommend publication of this manuscript after introduction of the following changes:

Comments:

1. Fig. 2. Negative stain EM of P3 VLP. Figure 2b should be included as an inset of Fig. 1a. Figure 1d should be removed as it adds no information; it is only a zoom of the central region of Fig. 1c.
2. p. 8, l. 164. Inclusion of an SDS-PAGE analysis of VLP in Fig. 2, showing the presence of VP0, VP1 and VP3, would support the sentence cited.
3. p. 8, l. 185-186. The sentence should be explained: Which suitable factors are referred to? How do the authors assert that sVLP are less stable?
4. Fig. 5. Relative to D and C antigenicity of rSktSC8 VLP compared to those of IPV, it seems that the C antigenic conformation is abundant in rSktSC8 VLP; a comment on this behavior should be made. I also wonder if the experiments in Fig. 5a were done with GPP3 bound to rSktSC8 VLP; would it be worth doing this to avoid conversion from D to C antigenicity? It is not clear why vSktSC8 VLP were analyzed in thermostability assays, but not in C or D antigenicity assays.

5. Include the indication “survival rate” at the top left of Fig. 6.

Reviewer #2 (Remarks to the Author):

OPV has flaws that include genetic instability, potential for genetic recombination, and lack of thermostability. IPV induces effective humoral immunity which protects against disease, but the IPV Salk vaccine is significantly more expensive than the Sabin attenuated, live vaccine. Production of both vaccines requires propagation of large amounts of infectious PV which carries the risk of an accidental reintroduction. The objective of the authors is to develop genetically stabilized PV virus-like particles. Here they demonstrate expression of PV VLPs of wt PV3 and the stabilized PV3 mutant SktSC8 as a cheap and viable source of vaccine to replace current IPV. The mutant retains the stability and potency of the current vaccine IPV and protects mice against challenge with wt PV. Structure determination of the PV3 SktSC8 sVLPs using cryo-electron microscopy (cryo-EM) shows that it adopts a native, D, antigenic conformation and analysis of the structure reveals the mechanism of action of the stabilizing mutations. The results are compelling and the manuscript is well written. Only a few minor issues:

Figure 3 please include the color key for the surface rendered map.

A central section of the map should be included to better illustrate map density and quality.

A few typos include “Coxsackievirus” (coxsackievirus) and “cryo-EM data was” (were).

Reviewer #3 (Remarks to the Author):

Marsian et al describe empty virus like poliovirus (PV) particles produced in plants through the expression of the P1 regions of either wt PV3 or the mutant PV3 SktSC8, identified as having stabilizing mutations within the coat proteins. P1 contains the genes for the capsid polyprotein of PV and the 3CD protease required for processing of the polyprotein into VP1, VP0, and VP3, the three proteins found in the particle prior to processing of VP0 into VP4 and VP2 that requires packaging of the RNA. They found that the 3CD protease had a detrimental effect on the plants when expressed at the same levels as the capsid polyprotein, reducing the production of the desired particles. They down regulated the expression of 3CD separately, reducing plant damage but maintaining sufficient quantities to process the polyprotein. They determined the high resolution structure of the stabilized SktSC8 with and without the GPP3 pocket factor mimic. The latter was determined to 3.6Å resolution and the former to 4.1Å. The structures were nearly identical to each other and to the authentic Sabine vaccine strain of PV previously determined.

There was some evidence of the pocket factor in the particles not treated with GPP3 indicating that the particles were finding a surrogate for this molecule in the plant cells. There was strong density for GPP3 in particles treated with this reagent and the binding was closely similar to that of the authentic pocket factor. They identified and rationalized the stabilizing influence of the 8 mutations in SktSC8 based on the structure. They then characterized the antigenicity of the particles. Wt PV3 produced C type antibodies, but produced D type antibodies when heat-treated. SktSC produced neutralizing D type antibodies without heat treatment. SktSC8 antibodies conferred protection in mice susceptible to PV infection that was comparable to previously established vaccine strains.

Overall this is a report of high significance. The production of structurally and antigenically authentic PV VLPs in plants is a first to this reviewer's knowledge. The analysis is complete at the structural and immunological levels and paves the way for commercial production of this VLP-based vaccine. Most importantly the SktSC8 mutant does not require a cold chain for activity, thus opening the way for a much less expensive path to PV immunity in third world countries.

The paper will be improved if the following points are addressed.

1. The introduction seems excessively long. Most readers of the paper will have an appreciation for the work without this amount of historical detail. I believe that the main points can be addressed in half the length.
2. It would be good to have a sentence or two in the main text generally describing the strategy for down regulating the production of 3CD protease relative to the polyprotein.
3. I am a bit confused about the structural comparisons between the SktSC8 and the Sabine strain PV particles. The Sabine strain is authentic virus and the VLPs produced here, as I understand it, have not undergone the final maturation. Yet they still have nearly identical structures with Sabine PV. Hogle determined the structure of empty PV (produced during a PV infection) that showed significantly different features from the PV nucleoprotein particles that had undergone VP0 to VP2 and VP4. The authors should make a few comments about how their particles look more like authentic PV than the "authentic" (ie. made during PV infection) empty PV.
4. In the same vein, when the authors are describing the locations of stabilizing mutations and their effect in SktSC8, they use VP4 and VP2 descriptors. Are they describing the regions of VP0 corresponding to these or have I misunderstood and these particles have undergone the final maturation. This should be clarified.

RESPONSE TO REVIEWERS' COMMENTS

REVIEWERS' COMMENTS:

Reviewer #1 (Remarks to the Author):

The manuscript "Plant-made polio 3 stabilized VLPs- a candidate synthetic polio vaccine" by Marsian et al. describes the effective assembly of virus-like particles of poliovirus 3 (PV3) with D antigenicity. These plant-expressed VLP were previously stabilized by introducing a few point mutations in the structural capsid proteins crucial for capsid integrity. Analyses by 3D cryo-EM of PV3 VLP at near-atomic resolution are impressive (taking into account the number of particles used for each map) and confirmed 7 of 8 amino acid substitutions (the remaining substitution is located in a disordered region). Finally, immunogenicity studies showed that these PV3 VLP elicit neutralizing anti-D PV3 antibodies. Adapted mice (in which a PV3 receptor is expressed) were immunized, and developed almost full protection to viral challenge. These findings are important for virus-based biotechnology and should be of interest to the structural biology community.

I recommend publication of this manuscript after introduction of the following changes:

Comments:

1. Fig. 2. Negative stain EM of P3 VLP. Figure 2b should be included as an inset of Fig. 1a. Figure 1d should be removed as it adds no information; it is only a zoom of the central region of Fig. 1c.

We assume that the reviewer is suggesting that Fig 2b should appear as an inset in Fig. 2a (not Fig 1a as stated) and that we should omit Fig. 2d (not 1d). On this Assumption, we have produced an appropriately modified version of Figure 2 with only 2 panels and an inset.

2. p. 8, l. 164. Inclusion of an SDS-PAGE analysis of VLP in Fig. 2, showing the presence of VP0, VP1 and VP3, would support the sentence cited.

We have actually analysed a particle preparation by SDS-PAGE followed by protein staining and have seen a pattern of bands consistent with the presence of VP0, VP1 and VP3. We have added a sentence to this effect, but are reluctant to go further as we have not actually gone through the formal identification of each product.

3. p. 8, l. 185-186. The sentence should be explained: Which suitable factors are referred to? How do the authors assert that sVLP are less stable?

The role and possible identities of factors capable of binding in the VP1 pocket are described in lines 87-93. The low electron density in the pocket of plant-made VLPs (lines 176-177) may be explained by the lack of suitable pocket factors in plant cells (lines 185-186) and we suggest (lines 186-187) that this may contribute to the lower thermostability of plant VLPs when compared to VLPs produced in mammalian cells which is apparent from data in Fig 5b. Text has been added to line 187 to clarify this. We have also replaced 'suitable factors' with the phrase 'factors suitable to substitute for the usual pockets factor(s).

4. Fig. 5. Relative to D and C antigenicity of rSktSC8 VLP compared to those of IPV, it seems that the C antigenic conformation is abundant in rSktSC8 VLP; a comment on this behavior should be made. I

also wonder if the experiments in Fig. 5a were done with GPP3 bound to rSktSC8 VLP; would it be worth doing this to avoid conversion from D to C antigenicity? It is not clear why vSktSC8 VLP were analyzed in thermostability assays, but not in C or D antigenicity assays.

We have modified the text to explain this and it now reads: “VLPs produced from the wt PV3 construct were entirely of non-protective C antigenic form whereas SktSC8 VLPs, **produced in plants or mammalian cells**, were predominantly in the native, D antigenic conformation (Figure 5a). **Some C antigen is present in SktSC8 VLP preparations, as it is in all virus preparations and IPV, probably reflecting <100% efficiency in the assembly process.**”

We also agree with the reviewer that it would be helpful to include C/D antigen analysis of vSktSC8 VLPs as a comparison. Fig5a has been modified accordingly and text included to comment on the C antigen content. GPP3 was not used in the experiments in Fig5a, which were carried out to analyse the effect of the stabilising mutations in the absence of any other factors.

5. Include the indication “survival rate” at the top left of Fig. 6.

A new version of Figure 6 has been produced to address this

Reviewer #2 (Remarks to the Author):

OPV has flaws that include genetic instability, potential for genetic recombination, and lack of thermostability. IPV induces effective humoral immunity which protects against disease, but the IPV Salk vaccine is significantly more expensive than the Sabin attenuated, live vaccine. Production of both vaccines requires propagation of large amounts of infectious PV which carries the risk of an accidental reintroduction. The objective of the authors is to develop genetically stabilized PV virus-like particles. Here they demonstrate expression of PV VLPs of wt PV3 and the stabilized PV3 mutant SktSC8 as a cheap and viable source of vaccine to replace current IPVs. The mutant retains the stability and potency of the current vaccine IPV and protects mice against challenge with wt PV. Structure determination of the PV3 SktSC8 sVLPs using cryo-electron microscopy (cryo-EM) shows that it adopts a native, D, antigenic conformation and analysis of the structure reveals the mechanism of

action of the stabilizing mutations. The results are compelling and the manuscript is well written. Only a few minor issues:

Figure 3 please include the color key for the surface rendered map.

This has been added.

A central section of the map should be included to better illustrate map density and quality.

We have done a resolution analysis and added this to the supplementary info

A few typos include “Coxsackievirus” (coxsackievirus) and “cryo-EM data was” (were).

We are happy to correct the typos identified by the reviewer, though we believe Coxsackievirus should be capitalised as it reflects the name of a town (Coxsackie in New York State).

Reviewer #3 (Remarks to the Author):

Marsian et al describe empty virus like poliovirus (PV) particles produced in plants through the expression of the P1 regions of either wt PV3 or the mutant PV3 SktSC8, identified as having stabilizing mutations within the coat proteins. P1 contains the genes for the capsid polyprotein of PV and the 3CD protease required for processing of the polyprotein into VP1, VP0, and VP3, the three proteins found in the particle prior to processing of VP0 into VP4 and VP2 that requires packaging of the RNA. They found that the 3CD protease had a detrimental effect on the plants when expressed at the same levels as the capsid polyprotein, reducing the production of the desired particles. They down regulated the expression of 3CD separately, reducing plant damage but maintaining sufficient quantities to process the polyprotein. They determined the high resolution structure of the stabilized SktSC8 with and without the GPP3 pocket factor mimic. The latter was determined to 3.6Å resolution and the former to 4.1Å. The structures were nearly identical to each other and to the authentic Sabine vaccine strain of PV previously determined. There was some evidence of the pocket factor in the particles not treated with GPP3 indicating that the particles were finding a surrogate for this molecule in the plant cells. There was strong density for GPP3 in particles treated with this reagent and the binding was closely similar to that of the authentic pocket factor. They identified and rationalized the stabilizing influence of the 8 mutations in SktSC8 based on the structure. They then characterized the antigenicity of the particles. Wt PV3 produced C type antibodies, but produced D type antibodies when heat-treated. SktSC produced neutralizing D type antibodies without heat treatment. SktSC8 antibodies conferred protection in mice susceptible to PV infection that was comparable to previously established vaccine strains.

Overall this is a report of high significance. The production of structurally and antigenically authentic PV VLPs in plants is a first to this reviewer's knowledge. The analysis is complete at the structural and immunological levels and paves the way for commercial production of this VLP-based vaccine. Most importantly the SktSC8 mutant does not require a cold chain for activity, thus opening the way for a much less expensive path to PV immunity in third world countries.

The paper will be improved if the following points are addressed.

1. The introduction seems excessively long. Most readers of the paper will have an appreciation for the work without this amount of historical detail. I believe that the main points can be addressed in half the length.

We agree that the Introduction was somewhat wordy in places and have trimmed it by about half a page. However we are reluctant to go further as we feel it would make the MS less accessible to a general readership.

2. It would be good to have a sentence or two in the main text generally describing the strategy for down regulating the production of 3CD protease relative to the polyprotein.

A sentence has been added to clarify this.

3. I am a bit confused about the structural comparisons between the SktSC8 and the Sabine strain PV particles. The Sabine strain is authentic virus and the VLPs produced here, as I understand it, have not undergone the final maturation. Yet they still have nearly identical structures with Sabine PV. Hogle determined the structure of empty PV (produced during a PV infection) that showed significantly different features from the PV nucleoprotein particles that had undergone VP0 to VP2 and VP4. The authors should make a few comments about how their particles look more like authentic PV than the “authentic” (ie. made during PV infection) empty PV.

The empty particles whose structure was determined by Hogle had undergone the D to C transition and therefore looked quite distinct from the native RNA-containing particles. The purpose of introducing stabilising mutations was to prevent this conversion, leading to the empty particles retaining the structure of the native (RNA-containing) particles. The cryo-EM structure shows that we have achieved this. We have added a sentence to the text: Such disorder is commonly seen in empty particles although these are usually in the expanded C antigenic form.

4. In the same vein, when the authors are describing the locations of stabilizing mutations and their effect in SktSC8, they use VP4 and VP2 descriptors. Are they describing the regions of VP0 corresponding to these or have I misunderstood and these particles have undergone the final maturation. This should be clarified.

The “cleaved” numbering is widely accepted even if cleavage of VP0 has not actually occurred. It has been used throughout the MS so that amino acids in the empties can be compared with those in the original virus. If we continued numbering through the whole of VP0, then the amino acid numbers downstream of the VP4-VP2 cleavage would be different in the two cases. However, as we now state, SDS-PAGE analysis is consistent with cleavage not occurring.